# Koji Starter and Koji World in Japan

**DOI:** 10.3390/jof7070569

**Published:** 2021-07-16

**Authors:** Hideyuki Yamashita

**Affiliations:** Higuchi Matsunosuke Shoten Co., Ltd., 1-14-2 Harima-cho, Abeno-ku, Osaka 545-0022, Japan; hi.yamashita@higuchi-m.co.jp; Tel.: +81-66621-8781; Fax: +81-66621-2550

**Keywords:** koji mold, koji starter, solid culture

## Abstract

Koji is made by culturing koji mold on grains. Koji has wide-ranging applications, for example, in alcoholic beverages and seasonings. The word ‘mold’ generally has a bad image, but in Japan, koji mold is valued for its usefulness, and over the years, efforts have been made to make safe, stable, and delicious food products from it. Koji mold spores, essential when making koji, are called koji starter in the industry. From the many available strains, those suitable for the production of each fermented food are chosen based on indicators such as growth rate and enzyme production capacity. In manufacturing using microorganisms, purity and yield are prioritized. However, the production of fermented foods using koji is more complex, with focus not only on the degree of decomposition of raw materials but also on factors influencing overall product design, including palatability, color, smell, and texture. Production can be facilitated by the variety of koji brought about by the diversity of koji mold combined with the solid culture method which increases the amount of enzyme production. In this report, we introduce the history of koji starter in Japan, the characteristics of koji mold in practice, and various fermented foods made from it. In addition, the factors affecting the quality of koji in solid culture are described.

## 1. Introduction

In Asia, especially East Asia, there are many foods fermented by molds that are thought to have been brought about by a hot and humid climate. In general, molds are considered toxic microorganisms for humans; however, the Japanese population has focused on the utility of koji mold and created unique koji-fermented foods and its food cultures since ancient times. There are wide variety of koji-fermented foods and liquors, such as sake (Japanese rice wine), honkaku shochu (a Japanese distilled liquor), amazake (a nonalcoholic sweet rice drink), soy sauce, miso (fermented soybean paste), mirin (sweet sake for seasoning), rice vinegar, salted koji, and pickles. In this way, koji mold represents a valuable resource that our ancestors carefully nurtured and used for a long time, and it has also contributed to our health and rich diet. Due to these reasons, the Scientific Conference of Brewing Society of Japan deemed koji fungi “the national fungi” in 2006 [1]. In addition, it was registered on the UNESCO (United Nations Educational, Scientific and Cultural Organization) Intangible Cultural Heritage list under the title “Washoku, traditional dietary cultures of the Japanese” in 2013 [2]. Many people all over the world have shown interest in the complicated umami that koji produces, as well as its aroma and flavors. Now, the flow of interest has returned to Japan. For some time, miso was not a common component of the standard Japanese diet because of its rich salt content, and amazake was regarded as an old-fashioned beverage; in recent years, however, these foods have been brought back into the spotlight, and both have experienced an unprecedented boom. While most Japanese people know and have eaten these koji-fermented foods, they do not know what koji is and that it is essential for the production of such foods. As koji mold is an invisible microorganism, it is difficult to imagine its shape from the final product (koji-fermented foods). Koji, which can be said to be the mystery of the East, plays a major role in common Japanese food. Naturally, even less is known about it in other areas of the world. Koji mold is recognized as safe because it has been eaten for a long time, and it is widely used as a host for the production of heterologous proteins and in the production of enzyme preparations using high enzyme production capacity. In addition, it is also attracting the attention of scholars as a tool for research into gene expression and secondary metabolites. In this review, koji mold used in the production of fermented foods is described.

First, this review briefly explains the difference between koji mold, koji starter, and “koji” in this text. Koji mold (kōji-kin in Japanese) is defined as “koji fungi”, the national fungi (partial revision by the change of taxonomical name…2015) [3].

The national fungi are:
(1)*Aspergillus oryzae*, “kikōji-kin” in Japanese.(2)*Aspergillus sojae* belonging to the *A. oryzae* group and their albino mutant strains.(Please note, *A. sojae* and *A. oryzae* are different species. This follows the taxonomy in the Japanese brewing industry and means that both belong to the group that forms yellow-green spores.)(3)Black *Aspergilli* group, namely *Aspergillus luchuensis* (*A. luchuensis* var. *awamori*), “kurokōji-kin” in Japanese, and its albino mutant *A. luchuensis* mut. *kawachii* (*A. kawachii*), “shirokōji-kin”.(Please note, *A. niger* is a different species from the black *Aspergilli* group, so it is excluded.)

Koji starter (“tane koji” or “moyashi” in Japanese) is the conidia (hereinafter called spores when used for industrial purposes) of koji mold formed on grains such as rice or barley, or spores harvested from it, the main component of which is the koji mold spore.

Koji is produced by inoculating koji starter onto a steamed cereal and culturing it while adjusting the temperature and humidity. It is called rice koji or barley koji.

## 2. Historical Background of Koji Starter and Koji

### 2.1. Generation of Koji and Koji Mold

What is the origin of koji mold? The answer to this question can be learned from the history of sake brewing. Since alcohol is the major component of sake and is obtained using carbohydrate as a substrate, it seems that various sakes were born from such materials. Because fruits such as grapes contain sugars that yeast can directly ferment into alcohol, it is inferred that alcoholic beverages came about by natural fermentation at an early time. On the other hand, sake brewing produced from wheat and rice may have started by discovering the saccharification of malt and germinated rice buds in the process of storing or cooking of grains. However, it seems that it took a long time to produce the method in an age when there was no koji starter, because koji was necessary for sake brewing using the koji enzyme for saccharification.

In *Qiming Yaoshu*, China’s oldest agricultural technology book written in the sixth century, it is recorded that koji or koji starter using steamed raw materials described as “yellow coat” is used for alcohol production. It is also noted that they were already used to produce “chi and jiàng” in Chinese, which is thought to be the origin of miso. In Japan, in the seventh century, it was written that sake was brewed from moldy rice, which is displayed on the home shrine. It seems that filamentous fungi grew naturally. This is believed to be the first record of sake brewing using rice koji produced without inoculation. On the basis of recent reports, it is reasonable to assume that koji mold exists in many of our surroundings, being transported by air and growing on rice over time [4]. Here, sake brewing began as a result of koji mold.

When using mold, koji and Chinese “qū”, Thai “luck pang”, Indonesian “ragi”, etc., the manufacturing methods are different. In the case of koji, the material is steamed, but in the other three, water is added to the crushed powder and used raw. Koji uses grains as they are, while the others are formed and used in various particle sizes. The used mold is different: koji is koji mold, but molds such as *Rhizopus* and *Mucor* spp. are used in the other cases [5]. These differences are reported in a comparative study with *A. oryzae* (RIB128) and *R. javanicus* (RIB5501) [6]. (1) In raw rice, RIB5501 grew faster than RIB128, but there was no substantial difference; (2) RIB128 with high acid carboxypeptidase activity (ACP) showed a minimal decrease in growth rate on steamed rice, but the growth of RIB5501 with low activity was significantly reduced; (3) since the growth rate of RIB5501 was improved by the addition of nitrogen compounds, the delay in growth on steamed rice seems to be due to the lack of supply of nitrogen sources. In the days when there was no pure culture technology for microorganisms, differences in the substrate and its processing method became a major factor in determining the mold to be used afterwards; thus, in Japan, where steamed materials were commonly used, koji mold grew more easily and seems to have been selected.

However, in *Engishiki*, the only valuable literature written about sake brewing in Japan in the 8th to 10th centuries, koji is produced by inoculation of a large amount of koji starter of 10% of koji weight (equivalent to 100 times the current amount of koji starter used), indicating the poor quality of the koji starter used at the time. In order to improve the quality of koji, focus was place on koji starter, and, consequently, an advance from natural inoculation to the tomo-dane (“friend of seed” in English) method was observed, in which seeds produced by keeping a good koji undisturbed for a long time were used. Regarding this, *Hoffmann* wrote that at the beginning of the sake production season, if *fermentreis* koji starter was not prepared, rice must be left in a room for a week or for an extra two to three days. During this time, spores of bacteria from the air adhere to the rice, and new growth of fermented bacteria is achieved. A notable point is that wood ash needs to be added to make mycelium grow as actively as possible, and it seems to work as a fertilizer [7]. The history of the use of wood ash in koji starter production is unknown, but it can be said that this is an epoch-producing technology that made it possible to manufacture stable koji starter at a time when sterilization technology was not developed. At a time when even the existence of koji mold could not be properly understood, this method continued until the late 1800s. During this time, repeated work with low reproducibility meant that progress in the koji starter production method was very slow.

At this time, koji starter was seldom sold, and each sake brewery produced their own. In the places where koji was continuously produced, it was used without drying because there was no need to preserve it. At the end of the sake-brewing season, lime and wood ash were mixed equally for the next koji production process, placed in wooden boxes, sealed, and stored. It was stored in a dry environment for long-term preservation, just as it is in modern day [8]. Advances in research into koji starter production methods and into koji mold were brought about by the above-mentioned European scholars who came to Japan after 1876. At this time, koji mold was separated and named in Japan [9]. For the researchers who knew of only malt as a glycolytic agent, koji, which produces a high concentration of 18% alcohol, was surprising and seems to have provided motivation for further research.

If “for sale” is added to the definition of koji starter, its history began in the Meiji era (1868–1912). It was sold in its naturally dried granular form mainly containing grain. In the 1900s, in addition to sake, there is also a record of koji starter being used for miso production. A special seed koji starter was sold, but there are some doubts about miso brewing. On the contrary, the use of commercial koji starter rather than tomo-dane was recommended, and the evaluation was not constant.

In addition, since the understanding of koji mold was ambiguous and sake was produced using only koji, it was assumed that koji was not only a source of koji mold spores but also of yeast thought to have originated from the hyphae of koji mold. Koji was valued as a source of yeast and lactic acid bacteria. At that time, koji mold was mainly composed of *Aspergillus* mold, but various microorganisms were mixed, and *Penicillium* was seen in the products of five of seven companies. It has also been reported that the number of bacteria and yeast was high, from 8000 to 300 million per gram, and the percentage of bacteria to spores was 0.23 to 2%, which indicates considerable contamination. This was the most serious issue of the tomo-dane method. In 1895, the importance of pure original strains used in koji starter production was proposed, and in 1911, a method for culturing them and for koji starter production using them was reported. Some koji starter manufacturers had already used this method, but it became mainstream after the Showa era (1926–1989). At the same time, koji starter was proved in 1926 to be a source of koji mold only, and further technical studies on pure cultures by koji starter manufacturers began. In 1907, three strains were isolated from sake koji starter for the first time, and since then, dozens of types have been separated from koji for sake, soy sauce, and miso, as well as products from various koji manufacturers; additionally, analysis of koji mold strains began. Since then, natural and artificial mutant strains have been added to this, and the number of strains has increased dramatically, with each company now possessing hundreds of koji strains.

### 2.2. Changes in the Classification of Koji Mold

The scientific name of koji mold, *Aspergillus*, first appeared in 1729. In 1876, H. Ahlburg was invited to the Japanese Medical College. He was the first to isolate koji mold from Japanese rice koji [10]. It was named *Eurotium oryzae* because it lacked the ability of sexual reproduction. In 1884, it was renamed *A. flavus*-*oryzae*. A major turning point in the discovery and classification of koji mold was the incident in 1960 involving the death of more than 100,000 turkeys in the U.K. The cause was peanut meal feed containing mold poison aflatoxins produced by *A. flavus* contamination [11]. Aflatoxins are secondary metabolites produced by certain strains in *Aspergillus* spp. and are known to be a cause of liver cancer. They are unavoidable, widespread natural contaminants of foods and feeds with serious impacts on health, agricultural and livestock productivity, and food safety; therefore, particular attention should be paid to aflatoxin contamination [12]. The fact that *A. oryzae* was in the same group as *A. flavus* according to the classification of C. Thom and K. B. Raper at that time, shook the Japanese brewing world [13]. This problem was thoroughly inspected by national and private organizations; all brewing koji molds were confirmed to be non-productive of aflatoxin, and concerns were resolved. In addition, multiple analysis methods revealed classification differences with an aflatoxin-producing strain and informed researchers both in and out of Japan that koji molds do not produce aflatoxin [14].

After that, the dispute over the classification of both strains continued. They were recognized to be the same species on the basis of 100% homology of their DNA [15]. In the following years, they were, however, distinguished via different electrophoresis patterns of DNA digestion by restriction enzymes, resulting in a different classification system between Japan and overseas [16]. However, in 2005, by genome sequencing and analysis of *A. oryzae* reported by the Japanese industry–university–government collaboration team, it was proven that *A. oryzae* and *A. flavus* are different species, and this long-standing issue was put to an end [17]. The analysis of the AF biosynthesis gene homolog cluster (AFHC) of koji mold has advanced, and the safety of industrial strains sold by Japanese koji starter manufacturers has also been confirmed. *A. oryzae* strains can be classified into three groups (group 1–3) based on the structure of AFHC. In group 1 strains, where AFHC is present, the expression level of the *aflR* gene is extremely low and there is no expression of *avn*A, *ver*B, *omt*A and *vbs*, which are necessary for aflatoxin production, and could not be confirmed by RT-PCR. Group 2 strains delete more than half of AFHC and group 3 strains amplify *vbs* at least. From these facts, it was found that *A. oryzae* is genetically very close to *A. flavus*, but does not produce AF [18,19]. By AFHC analysis of the *Aspergillus* strains for koji starter, it was confirmed that all of them were classified into group 1-3 and did not produce AF. It is unclear why koji molds have been selected in Japan since ancient times. It is suggested that *A. oryzae*, which lacks AFHC, is the mutant of *A. flavus*, and was domesticated by Japanese in the long history of brewing [20]. Alternatively, these mutants may have existed in Japan from the beginning. This report’s author would like to pay high respect to the wisdom of our forefathers, who selected and bred safe strains from among the many *Aspergillus* strains in nature. Subsequent reports on *A. sojae* NBRC 4239 [21,22] and *A. luchuensis* NBRC 4314 have also contributed to the confirmation of the safety of koji mold [23,24].

### 2.3. Current Koji Starter Industry

The history of using koji starter is long, but the current style of brewers purchasing koji starter from specialized manufacturers is a recent phenomenon. Previously, the number of brewers was large, but there was no technology to produce a large amount of koji starter. There were 43 companies belonging to the National Koji Starter Association in 1949, but now there are 12 companies. Among them, only six companies sell on a nationwide scale. Each company has a laboratory, which manages many strains; inspects product quality; carries out tests on koji production, breeds, and development; and conducts joint research with companies and universities. Koji starter has been manufactured throughout the year, and its production volume is highest for soy sauce followed by miso, shochu, and sake in terms of the amount of koji used.

In order to improve the taste and color of the final product, koji starter manufacturers continue to perform minor improvements to already produced products and search for new strains. After the first successful isolation in 1940, koji starter manufacturers acquired many strains from various materials. These characteristics are listed and used for screening as necessary. As the first example of breeding using artificial mutations by ultraviolet treatment, there is a deferriferrichrysin non-production strain that is a precursor to the coloring of sake. Recently, many products are sold for ginjo sake and, in response to the requests of brewers, the breeding of a white mutant for the lightening of food color is in progression. Regarding the mutation of koji mold, enzyme production using recombinant genes has been carried out, but it is not used in the field of brewed foods.

## 3. Industrial Koji Starter

### 3.1. Method for Producing Koji Starter

Koji mold germinates when an environment suitable for growth factors such as water, temperature, oxygen, and nutrients is prepared. It has a simple life cycle in which hyphae grow and form spores as they mature. It can be said that the production of koji starter is an extension of koji production, but the manufacturing process is quite different. While koji production is focused on obtaining enzymes necessary for the production of fermented foods, in the case of koji starter, the purpose is to obtain a large amount of spores that have a small number of bacteria and can be stored for a long time.

#### 3.1.1. Materials

Cereals such as rice, barley, and beans with a slightly polished surface are used to enable the growth of koji mold. The polishing degree allowing koji mold to grow and the spore yield increase is about 2–3% for rice and 10–20% for barley. The aim is to effectively provide nutrients, such as small amounts of minerals and protein, in the surface layer of grains that promote the growth of koji mold, and to prevent materials from sticking together when steaming. In general, since powdery materials have a large surface area, the spore yield increases, but there is also a disadvantage in that bacteria contamination can easily occur during koji production. Moreover, in strains with long hyphae, the spore yield may decrease because hyphae extend and become plate shaped.

#### 3.1.2. Moisture Content

The general cultivation time of koji is 40 to 45 h, but in the case of koji starter, it is longer, between 4 to 6 days depending on the strain. Koji mold consumes carbohydrates in the materials and generates heat; if left untreated, the product temperature exceeds 50 °C and the growth stops. In order to avoid this, it is necessary to lower the koji temperature. Since cooling of koji is performed using the latent heat of evaporation of water, the material moisture gradually decreases during cultivation. Moreover, when the koji moisture level drops to below 30%, the growth and spore formation of koji mold are suppressed. Consequently, it is important to maintain high material moisture and environmental humidity during cultivation to continue the growth of koji mold and to obtain high spore yields. Depending on the strain, a high material moisture of 45–50% can result in a large amount of spores, but under this condition, the risk of contamination by bacteria will increase rapidly, so the target value is about 40% in industrial production.

#### 3.1.3. Preparation of Koji Materials

First, sterilization of materials is conducted for the purpose of denaturation of starch and proteins suitable for the growth of koji mold. In order to completely sterilize heat-resistant spores such as *Bacillus* spp., pressurization steaming is desirable. What is important is the cooling process. Via sterile air through an HEPA (high-efficiency particulate air) filter, it is possible to maintain a nearly sterile state. The production of koji starter in Japan has the characteristic of the addition of about 0.1 to 0.5% wood ash when the material is cooled, and this contributed greatly to the priority cultivation of koji mold at a time when sterilization technology was not developed.

The effect of wood ash has been studied in detail, inluding: the growth-facilitating action of potassium phosphate, the antiseptic effect of the alkali degree of the ash, pH adjustment (neutralization of the acidic product), the physical effect (separation of rice grains from each other), and the nutritional effect. Trace components, such as copper, zinc, manganese, aluminum, etc., increase the growth of koji mold, promote spore formation, and contribute to the durability of the spore.

#### 3.1.4. Inoculation

Koji mold has a long logarithmic growth phase, so koji produced in the open system is easily contaminated with bacteria. The *Micrococcus* spp., which are the most commonly appearing bacteria in koji, are widely distributed in nature, such as in soil, and are universally found in food and food facilities. Moreover, they may cause food decomposition. In addition, since there are many cases where koji is made by hand, coliform organisms, which are considered to be hygienically defective, may be detected. *Bacillus* inhibits the growth of koji mold when the number of bacteria is high and reduces its quality. The count of heat-resistant spores increases rapidly in the case of temperatures above 35 °C, and they remain in foods, even in the presence of high salt, so special attention is required. Natto is a fermented food popular among people in Japan, but people involved in koji production refrain from eating it. Koji production is a battle against these bacteria. The optimum growth temperature of koji mold is 35 to 38 °C, and the optimum water content is about 40%, but this is a similar condition for other bacteria, so the cultivation of koji must start at a low temperature [25]. On the other hand, since the growth rate of koji mold is rapidly slowed below 25 °C, the aseptic cultured strain is inoculated at around 30 to 32 °C. Furthermore, since uniform inoculation of spores on the cooled material surface is effective for suppressing the bacterial count of koji, it is stirred many times after spraying.

#### 3.1.5. Culturing

The inoculated material is placed in a room at a temperature of about 30 °C, and the koji layer is thickened to 50–60 cm to prevent drying. At 20–24 h, white hyphae slightly cover the material surface, and koji temperature may rise to over 40 °C in strains with strong growth. Then, it is lightly mixed (teire in Japanese) to lower the temperature and provide oxygen, and subsequently thinly spread to a thickness of 1–2 cm in a wooden or metal container. In the case of culture using a wooden container (called koji-buta) (Figure 1a), koji cultivation continues in a room (koji muro) with a humidity of almost 100%, and moisture is adjusted by placing a sterilized wet cloth on the top. In this method, the cloth dried by the metabolic heat of koji is changed daily and moved from the top to the bottom of the room to reduce the temperature of the koji. This is an energy-efficient method, derived from traditional practice. However, this traditional method is performed in a humid environment and is suitable for koji mold, but since there is a hygiene problem, in recent years, a cultivation method by air conditioning management in a clean room has become mainstream.

After covering the material surface with hyphae, the stalk extends into the air and spores are formed at the top (Figure 1b,c). In order to harvest a lot of spores, mixing is not conducted after spreading. Strains with a long stalk and tightened into a plate because of the dense intertwining of hyphae have a white color and satisfactory appearance, but they also have poor spore formation, and the spore weight after drying is only about 3% of the material. In general, strains with a short stalk and fast spore formation, such as those for soy sauce production, have a large spore yield, and some strains exceed 10% per material. The point of the koji starter production process is to control the koji temperature in the second half at around 30 °C, which is the optimum temperature for spore formation without blowing or stirring.

#### 3.1.6. Drying

The moisture content of raw koji is from 32 to 35%, and that of spores is higher at about 40%. Matured spores usually have a high germination rate of over 90%, but when stored with high moisture, most spores die in about a month. The germination rate is calculated by microscopic observation of the number of germinations after liquid culture at 30 °C for 5 to 6 h. In order to enable long-term storage, koji starter is dried to under 10% of moisture content. The spores begin to die at temperatures above 45 °C, so drying is carried out in warm air dehumidified to 10–15% at 40–42 °C. Moreover, as spores are easy to scatter with their very light, fine powder characteristic, the blower must be as weak as possible, so the drying process takes 40 h. There is little decrease in germination rate while drying, but in cases of some strains it drops sharply to about 60%. Additionally, dried spores, like seeds of plants, easily absorb moisture and die rapidly under humid conditions. When dried koji starter is stored at a temperature of 5–15 °C and a humidity of under 50%, the germination rate is stable for about one year.

#### 3.1.7. Sieving

Most koji is sieved with 100–300 mesh, harvested, and preserved as spores. The appearance and enzyme activity are examined to confirm that there are no mutations during production.

#### 3.1.8. Preservation of Koji Strains

Koji mold is a valuable resource for koji starter manufacturers, so it is preserved by various methods, such as drying, liquid paraffin, glycerol, and freezing. The germination rate may decrease, but it is easy and effective to store at minus 20 °C in a sealed container. Usually, dried koji starter is stored at 15 °C and a humidity of 50%. In the case of an accident, they are stored in several places. Genetically unstable strains are also stored in different mediums. In order to avoid mutations, it is most effective to reduce the number of subcultures.

### 3.2. About Koji Starter

The form of the product is roughly divided into two granular types containing a medium such as rice and the powder type consisting of the spores of koji mold and starch. Currently, as koji-producing facilities have developed and continuous steaming and cooling are possible over several hours, powder types suitable for the inoculation device are mainly used, composing about 90% of all types.

There are only three species used in the brewing industry: *A. oryzae*, *A. sojae*, and *A. luchuensis*. About 40 strains are used for the production of koji starter. This diversity of koji strains is the reason why many practical koji starters are offered. There are many koji starters that are blends of multiple strains for the purpose of softening the too-strong characteristics or combining the desirable parts of both. In this way, the choice for brewers is wide.

The spores of *A. oryzae* and *A. sojae* are yellow-green to green, but their color slightly changes depending on the type of medium and culture time. In recent years, the use of white mutant strains has been increasing for the sale of koji and for the whitening of the color of miso and amazake. The spores of *A. luchuensis* are brown to black, but brown mutants have been used for a long time (Figure 2). The size of the spore is about 5 micrometers in diameter, and the number of dried spores varies by strain, from 5 to 30 billion spores per gram of koji starter. The general amount of inoculation in the case of grain is decided so that high-quality koji can be obtained based on the growth rate of strains and the number of spores. The granular and powder types are 1/1000th and 1/3000th–1/6000th of the material, respectively. Assuming that all spores adhere to the steamed rice, the number is calculated to be about 5000–10,000 per one grain.

The hidden quality improvement of koji starter is remarkable: the number of bacteria of koji starter was the same as that of contaminated koji, from 1 million to 100 million per gram in the early 1900s, but it has decreased sharply since around 1960, and in the 1990s, most powdery products had a purity of under 1000 per gram. Considering the above inoculation amount, it can be seen that there is no bacterium derived from koji starter on the material per gram at the start of koji production.

### 3.3. Fermented Food and Koji

While fermentation and decay are both the result of the living activity of a microorganism, the evaluation is divided on the basis of whether the product is beneficial for humans or not. When we add to the definition of fermentation that it can be controlled, it goes without saying that it is essential to use properly-managed microorganisms as starters rather than naturally occurring ones. The origin of starch-degrading enzymes in the production of alcoholic beverages using grains is roughly divided into two parts: one is malt in the dry climate, and the other is koji in the humid climate. The former represents the life activity of cereal, and the latter occurs due to the growth of koji mold on steamed rice without life.

#### 3.3.1. Brewing Industry in Japan

Currently, there are 1691 liquor manufacturers (1405 sake companies, 273 shochu companies, and 13 mirin companies). The production volume is about 366,000 KL of sake (20% alcohol equivalent), about 410,000 KL of shochu (25% alcohol equivalent), and about 93,000 KL of mirin. About 744,000 KL of soy sauce is produced by 1141 companies and 482,000 tons of miso is produced by 844 companies. These data are for 2019 from the Japan Sake and Shochu Makers Association, Soy Sauce Information Center, and the Japan Federation of Miso Manufacturers Cooperatives. Almost all companies purchase koji starter from six koji starter manufacturers.

#### 3.3.2. How Koji Works

Koji molds produce various enzymes, such as cellulase, protease, amylase, and lipase in order to degrade cereal consisting of complex tissues and to gain nutrients necessary for growth. The greatest feature of koji is that it has a complex of enzymes, so koji is said to be a treasure trove of enzymes. Amino acid and organic acid fermentations by microorganisms aim for the purpose of production efficiency of a single substance, but fermented foods produced by koji are evaluated on all degradation products. For example, glucose and amino acids produced by the enzymatic action of koji directly influence the taste and flavor of the product, and the amino carbonyl reaction that occurs during storage and coloring by oxidation affects the appearance of the product. These components are the nutrients for yeast and lactic acid bacteria and the substrate of alcohol and lactic acid fermentation; a small number of vitamins produced by koji are growth promotion factors. In addition, the mycelium of koji mold affects the texture of miso, and the space caused by hyphae breaking plays an important role in fermentation.

## 4. Koji Starter and Koji Suitable for the Production of Each Fermented Food

The manufacturing process of fermented foods produced by koji and the characteristics of koji mold strains used in such processes are shown (Figure 3). The quality of koji varies depending on the koji starter and koji production method, and the difference is reflected in the final product. *A. oryzae* is most widely used in all fields, *A. sojae* is mainly used in soy sauce, and *A. luchuensis* is mainly used in shochu. The production processes of each fermented food are described below.

### 4.1. Sake

Sake and shochu are produced by a parallel double fermentation method. Saccharification by koji and alcohol production by yeast occur at the same time; thus, the production method differs from that of beer and whiskey using malt, where each process occurs separately. Sake is produced from steamed rice, rice koji, and water, and it is then filtered. Most sake koji is produced using *A. oryzae*, which has high saccharification activity. However, if the rice is degraded too much, the off flavors increase and the sake quality degrades; therefore, koji should be made considering the balance between alcohol yield and taste. Some sakes are made using the fruit-like acidity of citric acid made by *A. luchuensis*.

### 4.2. Shochu

Shochu is known as a traditional Japanese distilled liquor made by the degradation of various carbohydrates sources, such as sweet potatoes, buckwheat, chestnuts, potatoes, carrots, sesame, and corn, with rice koji and barley koji. The rich amount of citric acid produced by *A. luchuensis* can suppress the growth of bacteria and secure the safety of its fermentation. *A. niger* also produces a considerable amount of citric acid, but it is thought to be impractical, because the flavor of koji and its products are not well-liked.

### 4.3. Soy Sauce

Soy sauce is a liquid seasoning fermented from koji with soybeans and wheat. “Tamari” is a dark-colored soy sauce produced with only soybeans, and “shiro shoyu” is white soy sauce produced with mainly wheat. Soybeans are treated under high pressure for the efficient consumption of their proteins. To prevent contamination by *Bacillus* spp., the moisture content of the surface is reduced with roasted and powdered wheat, and koji mold grows. To stimulate the utilization of proteins and to brew soy sauces with rich amino acids, especially glutamic acid, *A. oryzae* and *A. sojae* with high protease and glutaminase activity are selected and used for production. Strains with high amylase activity consume starch vigorously in koji production; thus, *A. sojae* with its low activity is useful for this purpose.

### 4.4. Miso

Miso is made by adding boiled soybeans and salt to rice koji or barley koji and fermenting it over several months. The color, taste, and flavor differ depending on the ratio of koji content, the fermentation temperature, and the fermentation period. Miso largely varies by region. The koji content in sweet miso, twice that in soybeans, is saccharified at 50 to 55 °C for several hours and has a low salt concentration of about 5%, and its color is light. Since the dissolution and degradation rate of miso proteins are approximately 60 and 25%, respectively, and it is a paste product, it is not necessary to increase nitrogen yield as in the case of soy sauce. If the degradation is too high, the miso becomes excessively soft, it changes color during storage, and its quality worsens.

### 4.5. Others

Mirin is a seasoning brewed with glutinous rice and rice koji and contains about 40–50% glucose. Amino acids are also involved in the taste, so koji with high saccharification and protease activity is necessary. Bright colors are preferred in its brewing, and strains with high tyrosinase activity are not appropriate. Amazake is a sweet beverage that is saccharified for several hours to 20 h at around 55 °C by adding water to rice koji. Koji with high saccharification enzyme activity is required for its production. Shio koji (salted koji) is a seasoning that has seen increasing popularity over the past decade. It is fermented at room temperature for 7 to 10 days by adding salt and water to koji. There are not as many sugars as in other seasonings, but the glucose, amino acids, and salt result in an effective condiment. Its most notable feature is that it is rich in active enzymes derived from koji, so meat soaked in it, for example, becomes tender and umami also increases. Strains with strong enzyme activity and a minimal Maillard reaction are required. Rice vinegar is produced by sake with acetic acid-producing bacteria. Unlike in the case of sake, all materials are decomposed using koji with high enzyme activity.

### 4.6. Diversity of Koji Mold

During the production of koji, the various characteristics of different strains, such as growth rate, metabolic heat, and length of hyphae; the sensory evaluation factors, such as aroma and taste of koji; and the enzyme production ability must be considered.

The growth rate up to about 20 h on rice indicates that there are differences depending on the koji mold species and strains: strains used for soy sauce shows slower growth than strains for rice koji, and *A. luchuensis* is even slower than both of these (Figure 4). From the appearance of soybean koji, it can be seen that the length of the hyphae and the color of the spores are different (Figure 5). Glucoamylase (GA) and acid protease (pH 3) or alpha-amylase activity in rice koji are correlated (Figure 6a,b); however, this activity has no relation to the growth rate of koji mold and the length of the hyphae.

Various substrates are fermented under the same conditions using one strain. Soy pulp (which is low in carbohydrates and rich in protein and dietary fiber) koji, adlay, and quinoa (which are difficult to utilize in the production of koji) koji, were compared, and the results showed that their protease (pH 6) and alpha-amylase (AA) activities differed by more than 10 times (Table 1).

Of course, the effect of the physical properties and the shape of the grain on the growth of koji mold is large, but it has been demonstrated that the large difference depending on the substrate is due to the complexity of the enzyme production mechanism of koji mold. In addition, in the experiment using wheat bran with starch removed, the large amount of starch-degrading enzyme activity of koji mold was extremely decreased, while other enzyme activity increased. In addition, the production of hemicellulase, which can degrade the hemicellulose of the major component of wheat bran, and cellulase enzymes, which had not previously been detected in wheat bran, has been confirmed [26]. It can be seen that the enzyme production of koji mold depends on substrate induction, and various responses are performed to adjust to the environment.

### 4.7. The Method of Koji Production

The culture method of microorganisms is broadly divided into liquid-state culture and solid-state culture. In the former, various nutritional ingredients can be used, and there are some advantages when compared with the latter in that uniformity by stirring is possible and temperature control is easy. However, it is recommended to adopt solid-state culture in the case of koji production not only because the substrate is grain but also because the glucoamylase gene (*glaB*), which plays a key role in the production of fermented foods, has been expressed specifically and highly in solid-state culture [27]. In addition, together with reports focused on acid proteases in rice koji, the mechanism of koji production, which developed over a long period of time, was elucidated at the gene expression level [28,29].

#### 4.7.1. Koji Production Process Differs Depending on Fermented Food

If rice is polished by more than 40% to remove proteins and lipids on the exterior, such as in the case of ginjo koji, koji production would be difficult, because rice is small and sticky. The normal koji production time is from 40 to 43 h, but it may exceed 50 h. In general, the ideal koji for sake brewing has high GA activity and GA/AA ratio, while that for ACP should be lower. However, in general, since there is a high correlation between the activity of these enzymes, this balance is lost in actual koji production. The first half of koji production suppresses the growth of koji mold, while in the second half, the temperature of koji is rapidly increased, and it is produced while being dried at a high temperature exceeding 40 °C.

There is a restriction that prevents strains with high GA from being used alone, because they also have high tyrosinase activity, which causes browning of sake cake (sake-kasu); thus, it is difficult to select the appropriate koji starter for sake brewing. Since there are various production methods for sake brewing and various types of sake, the number of strains used to respond to them is large, and there are many products with mixtures of strains.

The pattern of shochu koji production is unique; it starts from around 35 °C in order to improve the initial growth delay on the rice. Citric acid produced by *A. luchuensis* suppresses contamination by bacteria. Enzyme activity increases when the high temperature continues in the second half; however, citric acid production should be prioritized, and the temperature of koji is gradually lowered to 30–35 °C in the latter half (Figure 7a).

In soy sauce, the temperature of koji is controlled to increase protease activity. Due to the characteristics of the material, it can easily be contaminated by bacteria, so the initial temperature of koji is 30 to 32 °C. In addition, due to the high metabolic heat, it is mixed earlier than the other forms of koji to lower the temperature, and after 24 h, it is controlled between 25 and 28 °C (Figure 7b). The protease activity (pH 6) of soy sauce koji is considerably higher at approximately 10 times that of miso koji.

Since miso is a paste seasoning, the shape of koji directly affects the appearance of the product; koji covered with mycelium firmly on the surface of the grain is required. Although amino acid production is essential, glucose to soften the salty taste is also required, and high temperatures and a drying process that limits enzyme activity as with sake are not used (Figure 7c). In the case of soybean koji, a unique temperature process with the highest emphasis on bacillus suppression is used (Figure 7d).

Koji for salted koji (shio koji) and amazake is essentially produced in the same way, but strains with low tyrosinase activity are often used to avoid coloring the product. The tyrosinase activity of koji is low in salt tolerance, and miso is colored by the amino–carbonyl reaction, so it is necessary to pay attention to this activity as in the case of sake and amazake.

#### 4.7.2. Prevention of Bacteria Contamination

Koji production is a fight against bacteria contamination because the environment for producing koji is also suitable for bacteria. Since koji production is performed in an open system, the bacteria adhered in the process increase from 1000 to 100,000 times during the koji production process. Temperature control of up to several hundred Kgs is often carried out manually for up-scaled production. Cooling of koji temperature with a blower is adopted for proper temperature control. The point of the solid-state culture of koji is to remove metabolic heat and CO_2_ generated with the growth of koji mold and to achieve the target temperature. However, it is considerably difficult to uniformly control the koji temperature due to differences in the growth of koji mold as a result of the irregular shape of the grain and the place in the machine. However, recently, there have been some remarkable improvements in the technology of brewing facility manufacturers. The control of the whole process, from material preparation to koji production, is fully automated. The machine is larger, and it can produce 30 tons for rice koji and barley koji and 70 tons for soy sauce koji with one machine. In addition, nowadays, automatic cleaning and sterilization of the facility are available. The development of the facility for aseptic culture is also progressing every day.

## 5. Recent Topics on the Use of Koji Mold and Koji

Until recently, koji has mainly been used for fermented foods produced from cereals, but there are some other unique developments. These are the results of a combination of ingenuity in material processing and the vitality of koji mold adapted to various environments, and there is much potential for different uses of koji mold in the future.

### 5.1. Ocha (Tea) Koji

Chinese pu’er tea is a fermented tea. Natural fermentation is performed for its production, but it is difficult to verify the usefulness of microorganisms because the fermentation period is long and the microflora changes. It was found that tea with high reproducibility was produced by short-term fermentation with koji mold, and the astringency of the tea was reduced while its umami was enhanced [30]. The degradation of catechins was confirmed, and, as a result, much attention has been paid to the possibility of functional components produced by koji mold.

### 5.2. Seaweed Koji

A seaweed that is popular among the Japanese population contains abundant nutrients and active elements; however, due to its thick cell wall, it is poorly digested, and it cannot be said to be effectively utilized. Since poor-quality seaweed is not suitable as food, most are composted or disposed of, but a more effective and efficient use of them is necessary. Therefore, for the purpose of using seaweed with koji mold, koji was produced using *Neopyropia yezoensis*, *Saccharina japonica*, and *Sargassum horneri*. Since seaweed has high moisture content and it is difficult to make koji as it is, they were dried and powdered to promote the growth of koji mold. Seaweed koji contained enzymes that break down on their own, such as proteases, amylases, and mannanase. By using it in the production of seaweed soy sauce, liquefaction and degradation were greatly promoted, and the brewing period was shortened by half. Seaweed sauce is allergen free and richer in flavor than soy sauce, and koji successfully added depth to its taste [31].

### 5.3. Egg Koji

Eggs are foods that contain proteins and high-quality lipids and are nutritious, but they are mainly heated for eating, so most of the nutrients are wasted. It was difficult to directly ferment eggs with koji because of the specificity of this material and bacteria contamination of the egg, but it became possible by devising a method for this processing. Egg koji has an enzyme composition of almost no amylase and high protease activity, and if it is aged with egg yolks, the total amount of amino acids increases by about five times compared to that of general egg yolks. In addition, not only did the aroma of egg yolks significantly increase, but a new fragrance was also produced, indicating the usefulness of egg koji [32].

### 5.4. Koji Cheese

Cheese is a universally loved food produced from dairy products; some cheeses are fermented with mold. There are various types, such as Camembert cheese using *Penicillium camemberti and P. candidum* and blue cheese using *Penicillium roqueforti*.

Some koji molds used for surface mold-ripened cheeses, such as *P. candidum*, have protease activities but significantly lower lipase activity. Since the cheese produced with this strain showed a mild flavor due to a decrease in volatile short-chain fatty acids; it is expected to promote a new trend in the field of surface mold-ripened cheeses [33].

### 5.5. Liquid Culture of Koji Mold

The reason why the liquid culture method of koji has not been adopted in alcoholic production is that in *A. oryzae*, the most important GA activity in alcohol fermentation is lower than that with solid-state culture, and in the production of shochu, acid resistant AA of *A. luchuensis* is not produced in liquid-state culture. To solve this issue, a method of using a liquid medium to which cereals are added was devised [34]. Liquid koji with GA and high acid resistant AA activity were cultured using liquid koji starter. This is an epoch-producing method that enables the production of liquid culture of koji mold, which has been difficult to achieve until recently.

## 6. Conclusions

Advances in genetic analysis of koji mold have resulted in a large amount of information on the safety of koji mold, enzymes, and metabolites and have greatly contributed to research and practical use. In addition, due to the rapid improvement of gene editing technology in recent years, research on koji mold has entered a new stage. In terms of practical use, the production method of fermenting foods using Japanese koji is complete, so gradual steps will continue in the future. However, looking overseas, it is expected that new fermented foods using koji mold will be created, because tastes and preferences differ from those in Japan, and there are many crops and cereals that do not exist in Japan. In addition, there is a possibility that a new encounter between koji mold and various materials will create useful substances. For those who study koji mold, I hope that this review will be an opportunity to learn more about koji starter, koji, and koji mold itself. Thank you for giving me this opportunity to report from the standpoint of a koji manufacturer.

## Figures and Tables

**Figure 1 jof-07-00569-f001:**
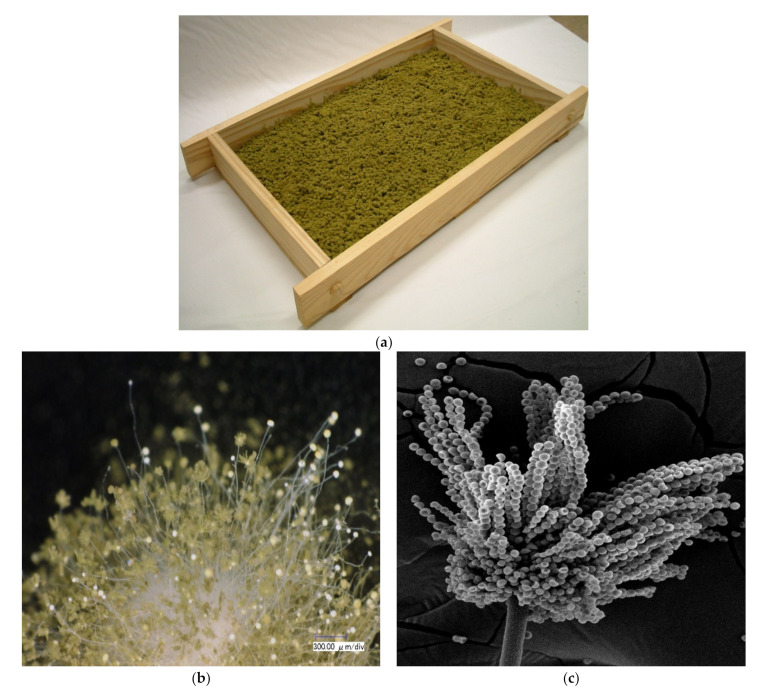
Koji starter and koji: (**a**) koji starter for sake produced by traditional method; (**b**) the appearance of koji mold (×400: photo is taken by SEM (Scanning Electron Microscope) TM3030Plus made by Hitachi High-technologies Corp.); (**c**) conidia formation of *Aspergillus oryzae* W-20 (96 h).

**Figure 2 jof-07-00569-f002:**
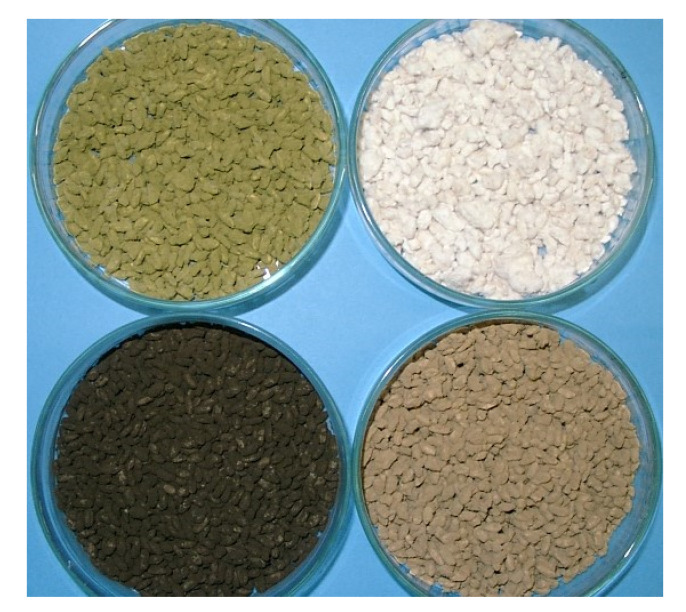
Color mutant of koji mold: upper image—*Aspergillus oryzae*; lower image—*Aspergillus luchuensis*.

**Figure 3 jof-07-00569-f003:**
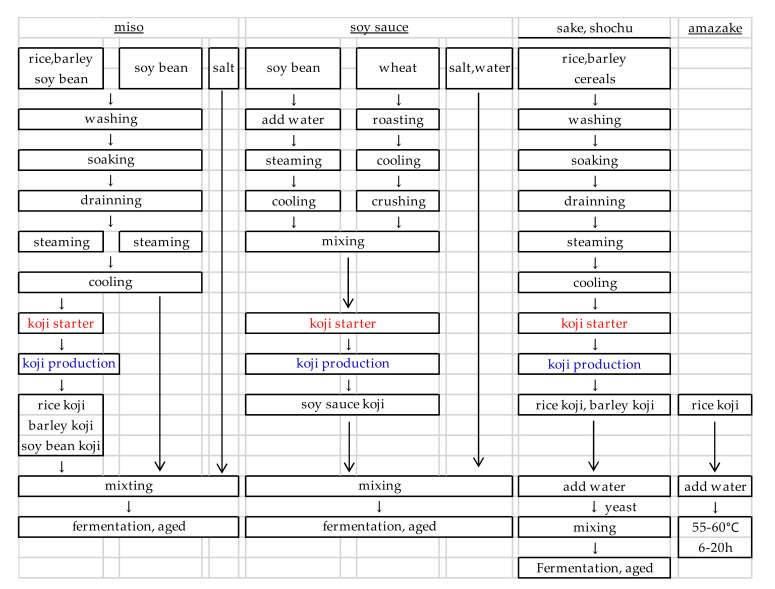
The manufacturing process of fermented foods.

**Figure 4 jof-07-00569-f004:**
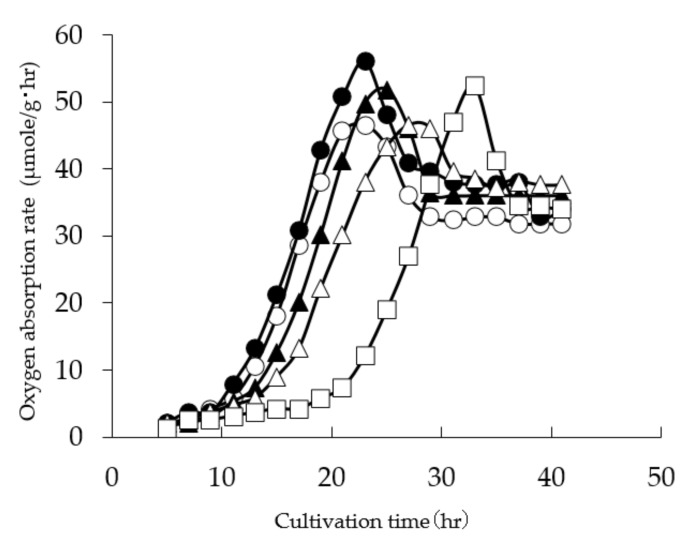
Differences in growth rate on steamed rice: (-●-) *Aspergillus oryzae* W-20; (-○-) *Aspergillus oryzae* W-52; (-▲-) *Aspergillus oryzae* S-03; (-△-) *Aspergillus sojae* No.9; (-□-) *Aspergillus luchuensis* SH35; hr hour.

**Figure 5 jof-07-00569-f005:**
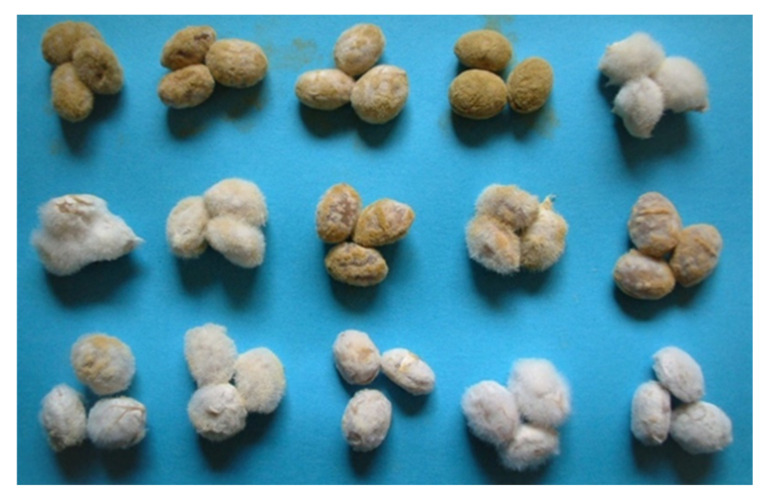
Soybean koji made with different koji strains.

**Figure 6 jof-07-00569-f006:**
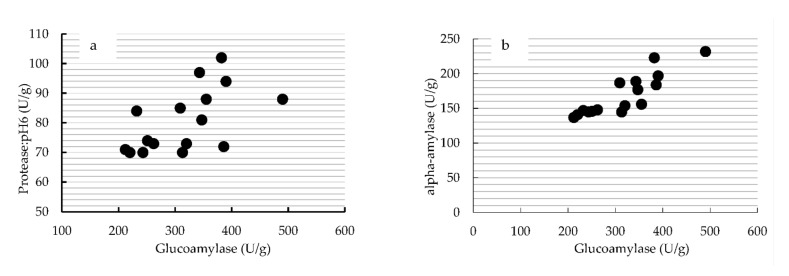
Correlation of enzyme activity (U/g) produced by koji starter for miso: (**a**) glucoamylase and protease (pH 6); (**b**) glucoamylase and alpha-amylase.

**Figure 7 jof-07-00569-f007:**
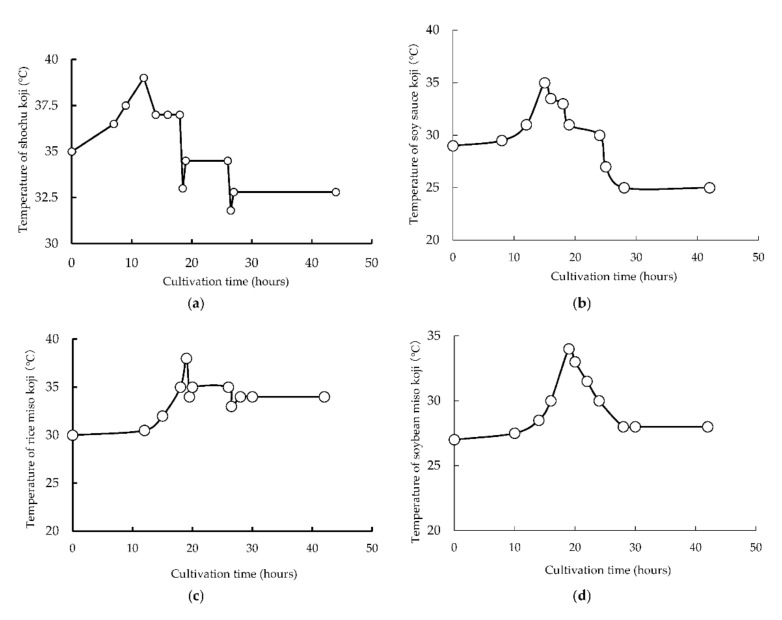
Production process (-○-) of koji suitable for each fermented food example: (**a**) shochu; (**b**) soy sauce; (**c**) rice miso; (**d**) soybean miso.

**Table 1 jof-07-00569-t001:** Enzyme activity of various substrates’ koji made with *Aspergillus oryzae* W-52 (for industrial use). Value when enzyme activity of adlay is 1.0.

	Glucoamylase	Alpha-Amylase	Protease (pH 3)	Protease (pH 6)
Millet	1.6	2.1	3.5	2.5
Germ of rice	1.9	4.5	2.6	4.4
Barley	1.4	1.7	1.7	1.5
Adlay	1.0	1.0	1.0	1.0
Brown rice	1.7	2.2	1.9	1.4
Buckwheat	1.4	1.4	2.1	1.6
Soybean	1.7	2.6	0.4	2.1
Rice protein	2.2	15.1	0.2	5.7
Soy pulp	2.6	13.7	5.8	11.7
Quinoa	1.3	1.5	1.0	0.9

## Data Availability

The data are available within the manuscript.

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
