# Peer review of "Koji Starter and Koji World in Japan"

_jof, 2021, doi:10.3390/jof7070569_

Round 1

Reviewer 1 Report

The article "Koji starter and koji world in Japan" is interesting and worth publishing. The author of this manuscript is a practitioner of koji production on a daily basis, and his approach to the subject may be interesting for scientists. The style used by the author (singular / plural first person form instead of the impersonal form) slightly differs from the typical style of publication in scientific journals, but in my opinion it does not reduce the value of this work.

I would suggest making a few changes to the text that should improve its overall quality. As I am not a native English speaker, some of the suggestions concerning the style/the usage of tenses should be further consulted with a professional language translator.

I would like to ask the author to comment on the questions included in the following comments:

Page 2 line 92-93. The sentence: ‘Koji uses (…) in various sizes’. As I understand, it is about the difference in the substrate particle size. I would suggest using ‘in various particle sizes’ instead of ‘in various sizes’ in this sentence.

Page 2 line 96- ‘(RIB5501) was faster’ should be replaced with ‘grew faster’

Page 3 line 98- ‘but the growth of RIB5501’ ?

Page 3 lines 113-123 – in this passage, in lines following ‘… Hoffmann wrote...’ I would suggest using past tense (‘koji starter was’, etc.) instead of present tense. Moreover, in line 116, ‘fermented bacteria’ should be replaced with ‘fermenting bacteria’.

Page 3 line 139- ‘ the use’ or ‘the usage of’?

Page 3 line 143- the fragment ‘yeast made from the hyphae’ is incomprehensible for me. Could you please explain it? If this phrase shows the state of knowledge about koji in Japan in the 1900s, this should be clearly indicated, otherwise it is confusing for the reader.

Page 3 line 144- 145, following ‘koji mold was mainly composed of mold’ - Penicillium is also a kind of mold. A more precise term would work better here, in order to clarify the text, for example ‘... of Aspergillus mold’.

Page 3 line 147 ‘the number of bacteria containing yeast’ is incomprehensible. I would suggest changing this into ‘the number of bacteria and yeast’ or ‘combined numbers of bacteria and  yeast’.

Page 3 line 150-151 in‘ culturing them and for koji starter production using it was reported.’- ‘ using them...’ seems better?

Page 4 line 163- ‘It is named’ - past tense should be used instead

Page 4, line 171 - ‘… particular attention should be paid to [12].’ – to... ?? this sentence lacks the ending

Page 4 line 171- 173- maybe this sentence should start with ‘The fact that (…) according to the classification (…), shook the Japanese (…)’

Page 4 line 175-176- please delete ‘the changed various culture condition’ from the sentence (this part of the sentence is not necessary, and furthermore, it may be incomprehensible)

Page 4 line 179-182- please re-edit these two sentences as they seem controversial. Maybe: instead of ‘- their DNA (…)’, it would be better to write : ‘they were recognized to be the same species on the basis of 100% homology of their DNA. In the following years, they were however distinguished via (…)’  or something like that.

Page 4 line 186- I think it should be ‘biosynthesis’ rather than ‘biosynthetic’

Page 5 line 221- ‘(except koji mold)’- this text in the parenthesis seems not necessary in this sentence

Page 5 line 227- ‘provide’ instead of ‘use’ would be a better word choice; also, the word ‘contained’ is not necessary

Page 6 line 268- high or big instead of ‘much’

Page 6 line 271- people in Japan

Page 6 line 299-300- please re-write this sentence to improve its style; also – ‘soy sauce production’ would be better

Page 7 line 345-instead of ‘it’ should be ‘koji/koji starter’?

Page 4 line 350- ‘but in cases of some strains it drops sharply’ would sound better here

Page 8 line 357 –‘of multiple strains’?

Page 8 line 384- per gram of what? Koji starter?

Page 9 line 420- ‘a complex of enzymes’?

Page 9 line 421-423- The sentence which starts with ‘Amino acid (…)’ is not clear, could you please improve it?

Page 9 line 424- instead of ‘directly become, ‘directly influence’ would be better

Page 9 line 426-428. This sentence is unclear: ‘These components are the nutrients for? yeast and lactic acid bacteria and the substrate of alcohol and lactic acid fermentation? , and? a small (…).’

Page 11 line 484- ‘various carbohydrates sources?’

Page 12 line 538-542. First of all, what does ‘cereal’ mean in this sentence? In table 1, there are various substrates for koji, not only cereal- soy bean, soy pulp, buckwheat and quinoa. Furthermore, the description of enzyme activities is not consistent with results presented in table 1. Moreover, a grammatical error has crept into this paragraph- if ‘one strain’ so why ‘their’ enzyme activity?

Page 14 line 596- ‘reports focused on ? acid protease’

Page 14 line 597- ‘… the mechanism of the koji production method…’ - I would suggest simplifying this fragment, maybe the word ‘method’ can be removed?

Page 14 line 603- ‘should have high’ instead of ‘has high’?

Page 15- the description of y axis in Figure c is incorrect

Page 16 line 692- ‘A seaweed that is? popular’

Page 16 line 702 ‘Seaweed sauce is a cereal’ -it sounds very, very strange. What do you mean?

Page 16 line 708- ‘a method of this? material’

Page 16 line 708- ‘an enzyme composition specialized in protease activity’ should be re-phrased in order to make the phrase more clear

Page 16 line 714- there is a generalization in this sentence, because cheese obtained after fermentation with molds are only part of cheese production.

Page 16 line 717- P. candidum is not a correct example? Probably, some koji mold strain should be listed here.

Additionally, I would like to ask the author to check references and mark in proper position if a publication is available only in Japanese. For example, number 32 lacks this indication.

Author Response

Dear sir

Thank you for your polite opinions.

They were very helpful for me.

Please see the attachment for revisions based on your suggestions.

There was only one unknown part (page 14 line 603-should have high),

where is this part?

Sincerely yours. 

Hideyuki Yamashita

Reviewer 2 Report

This manuscript introduced the history of koji starter in Japan, the characteristics of koji mold and various fermented foods which made from it. In addition, the factors affecting the quality of koji in solid culture are well described. However, the description of modern technology and frontier development of Koji Starter and Koji is few in this review, in order to get close to “Journal of Fungi”, some points should be considered as follows.

  1. The safety of Aspergillus oryzae (koji mold) is not described clearly in this manuscript. It does not well reveal the real differences between A. flavus and A. oryzae. The differences between A. flavus and A. oryzae should be analyzed from the perspective of genomics. (For example, the whole genome sequencing showed that the reason why A. flavus produce aflatoxin is for the existence of genes cluster encoding the aflatoxin biosynthesis, which do not exist in the genome sequence of A. oryzae. In my opinion, in order to make this manuscript elaborate and scientific, the genes name, sequence length which encoding aflatoxin biosynthesis in A. flavus, but not exist in A. oryzae should be added to the section of “Changes in the Classification of Koji Mold” in this manuscript.

  1. The relationship between A. flavus and A. oryzae are not well described in this review. A. flavus and A. oryzae belong to the same genus and different species. It is suggested that A. oryzae which lack the gene cluster encoding the aflatoxin biosynthesis, is the mutant of A. flavus in the long evolution, and was screened the by Japanese in the producing process. In my opinion, this part should be added to the section of” Changes in the Classification of Koji Mold” to make the relationship of A. flavus and A. oryzae clear.

Author Response

Dear sir.

Thank you for your advices.

As you said, I added it in the text, so please refer attachment.

Sincerely yours.

Hideyuki Yamashita

Reviewer 3 Report

This review introduces the history of koji mold in Japan, the characteristics of koji mold actually used, and various fermented foods using koji mold. In addition, factors affecting the quality of koji in solid-state culture methods are discussed. This is very well organized and each item is carefully summarized, so this is an excellent review that can be learned by both beginners and experts who are interested in koji.    

L56, Aspergillus sojae belonging to the A. oryzae group and their albino mutant strains.

What ‘group’ means? Word in taxonomy? Maybe better more carefully here and add reference.

L57-59, citations for A.luchuensis and kawachii are helpful.  I found citations about them later, L189-191. My suggestion is, L60 (see below, please note, A.niger is a different species,,,,,)   Fig.4, 5. Are there any citations on the experiments? If they are new data, we need materials and methods for them, but it does not fit to the review style.   Fig.7. Are these continuous measurement of the temperature of koji? Is the temperature of the room or chamber used to control the temperature of koji? Is there any citation on the experiments?  

Author Response

Dear sir

Thank you for your advices. As you said I added it in the text, so refer attachment.

About Fig.4, 5, 

The initial moisture content and koji production process differ depending on the materials used in each fermented foods, and further fine-tuned by each manufacturing company, so there is no citation on the experiment. 

About Fig. 7,

The continuous blot shows the temperature of koji. As koji mold grows, it generates heat, so koji temperature is controlled by the room temperature and humidity of the environment.

Sincerely yours

Hdeyuki Yamashita
